# Nonequilibrium fluctuations of a quantum heat engine

Tobias Denzler,[1] Jonas F. G. Santos,[2, 3] Eric Lutz,[1] and Roberto M. Serra[2]

[1]*Institute for Theoretical Physics I, University of Stuttgart, D-70550 Stuttgart, Germany*
[2]*Centro de Ciências Naturais e Humanas, Universidade Federal do ABC,*
*Avenida dos Estados 5001, 09210-580 Santo André, São Paulo, Brazil*
[3]*Faculdade de Ciências Exatas e Tecnologia, Universidade Federal da Grande Dourados,*
*Caixa Postal 364, Dourados, CEP 79804-970, MS, Brazil*

The thermodynamic properties of quantum heat engines are stochastic owing to the presence of thermal and quantum fluctuations. We here experimentally investigate the efficiency and nonequilibrium entropy production statistics of a spin-1/2 quantum Otto cycle in a nuclear magnetic resonance setup. We first study the correlations between work and heat within a cycle by extracting their joint distribution for different driving times. We show that near perfect correlation, corresponding to the tight-coupling condition between work and heat, can be achieved. In this limit, the reconstructed efficiency distribution is peaked at the deterministic thermodynamic efficiency, and fluctuations are strongly suppressed. We further successfully test the second law in the form of a joint fluctuation relation for work and heat in the quantum cycle. Our results characterize the statistical features of a small-scale thermal machine in the quantum domain, and provide means to control them.

## 1 Introduction

Heat engines have played a prominent role in our society since the industrial revolution. They are commonly used to generate motion by converting thermal energy into mechanical work [1]. An important figure of merit of heat engines is their efficiency, defined as the ratio of work output and heat input. According to the second law of thermodynamics, the maximum efficiency of any thermal motor operating between two heat baths is given by the Carnot formula, $\eta_{ca} = 1 - T_1/T_2$, where $T_{1,2}$ denote the respective temperatures of the cold and hot reservoirs [1]. Standard heat engines, such as car engines, usually operate in a regime where energy fluctuations are much smaller than mean energies. As a consequence, heat, work and, hence, efficiency are deterministic quantities.

In the past decade, successful miniaturization has led to the experimental downscaling of thermal machines to microscopic [2–4] and nanoscopic [6–10] levels. Quantum heat engine operation has furthermore been reported recently in a variety of systems [11–15]. Such devices are typically subjected to thermal [16, 17] and, at low enough temperatures, to additional quantum [18, 19] fluctuations. These are associated with random transitions between discrete energy levels, and thus introduce nonclassical features. As a result, heat, work, efficiency, and other relevant thermodynamic quantities such as the nonequilibrium entropy production, are stochastic variables. These fluctuations strongly impact the performance of microscopic and nanoscopic machines [20–23]. Understanding their random properties is therefore essential. The efficiency statistics of classical Brownian heat engines has been studied experimentally with optically trapped colloidal particles in Ref. [4]. Remarkably, efficiency fluctuations above the Carnot efficiency, which originate from negative entropy production events, have been observed [4]. Meanwhile, the random entropy production for arbitrary heat engines has been theoretically predicted to satisfy a fluctuation relation [24–27], a fundamental nonequilibrium generalization of the second law of thermodynamics for small systems [17–19]. Such a fluctuation theorem has recently been experimentally simulated with a quantum computer for a quantum swap engine [28]. However, the efficiency and nonequilibrium entropy production statistics of a general quantum heat engine have not been explored experimentally so far.

We here report the study of the fluctuating properties of a proof-of-principle quantum Otto engine [29] based on a driven nuclear spin-1/2 in a liquid state nuclear magnetic resonance (NMR) setup [30]. We extend existing interferometric methods [31–34] to experimentally extract the joint distribution of work and heat for different cycle times. This joint distribution is crucial for the detailed analysis of energy fluctuations in cyclic processes, including the stochastic efficiency and the joint fluctuation relation for work and heat. We first exploit the multipoint statistics to investigate the correlations between work and heat during an engine cycle, from the adiabatic to the nonadiabatic regime. We find near perfect correlation, corresponding to the tight-coupling condition between the possible values of work and heat [35–38], in the quasiadiabatic limit. We additionally determine the distribution of the quantum stochastic efficiency and analyze the impact of the work-heat correlations on its features. We show, in particular, that, as the tight-coupling regime is approached, the random efficiency is peaked around the deterministic (thermodynamic) efficiency, and its fluctuations are strongly suppressed. We finally verify both a detailed and an integral bivariate quantum fluctuation relation for cyclic heat engine operation [24–27] and examine irreversible losses associated with quantum friction [39–41].

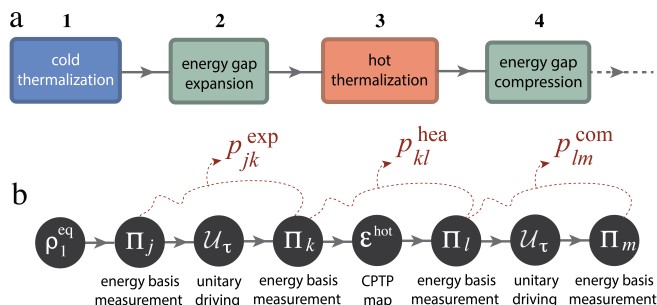

FIG. 1. Quantum heat engine. (a) Four steps (cooling, unitary expansion, heating, unitary compression) of the quantum Otto cycle realized in the experiment. (b) Multi-point-measurement scheme used to determine the joint distribution $P(W, Q)$ of work and heat: projective energy measurements are performed at the beginning ($\Pi_j$) and at the end ($\Pi_k$) of the expansion stroke, as well as at the beginning ($\Pi_l$) and at the end ($\Pi_m$) of the compression phase. Each pair of measurements is realized via a Ramsey-like interferometric method. The operator $\mathcal{U}_\tau$ describes unitary driving and $\varepsilon^{\mathrm{hot}}$ characterizes the completely positive trace preserving (CPTP) map that fully thermalizes the system to the hot temperature.

## 2 Experimental system

In our experiment, we use a $^{13}$C-labeled CHCl$_3$ liquid sample diluted in Acetone-D6 and a 500 MHz Varian NMR spectrometer. We employ the spin 1/2 of the $^{13}$C nucleus as the working medium of the quantum engine and the $^{1}$H nuclei as a heat bus to deliver heat to the machine. Work is performed by driving the engine with a resonant radio-frequency (rf) field. The low rf modes near to carbon resonance frequency will effectively play the role of the cold bath, while high rf modes near the hydrogen Larmor frequency that of the hot reservoir.

We realize a quantum Otto cycle that consists of four different steps [29] (Fig. 1a). 1) Cooling: the $^{13}$C nuclear spin is initially cooled, using spatial average techniques [30], to a pseudo-thermal state $\rho_1^{\mathrm{eq}} = \exp(-\beta_1 H_1^{\mathrm{C}})/Z_1$ at cold inverse spin temperature $\beta_1$, where $H_1^{\mathrm{C}} = -(h\nu_1/2)\sigma_x$ is the initial Hamiltonian and $Z_1$ is the partition function. 2) Expansion: the machine is then driven by a time-modulated rf field on resonance with the $^{13}$C nuclear spin. In a rotating frame at the $^{13}$C Larmor frequency ($\approx 125$ MHz), the driving is described by the following effective Hamiltonian, $H_{\mathrm{exp}}^{\mathrm{C}}(t) = -(h\nu(t)/2)\left[\cos(\pi t/2\tau)\sigma_x^{\mathrm{C}} + \sin(\pi t/2\tau)\sigma_y^{\mathrm{C}}\right]$, where the nuclear spin energy gap, $h\nu(t) = h\nu_1(1 - t/\tau) + h\nu_2 t/\tau$ is varied linearly from $\nu_1 = 2.0$ kHz at time $t = 0$ to $\nu_2 = 3.6$ kHz at time $t = \tau$, where $\sigma_{x,y,z}^{\mathrm{C}}$ are the Pauli spin operators of the $^{13}$C nuclear spin. Implemented driving times ($\approx 10^{-4}$ s) are much shorter than the typical decoherence times in our setup (few seconds). The system is hence isolated from its surroundings to an excellent approximation and the corresponding evolution $\mathcal{U}_\tau$ is unitary [33]. 3) Heating: heat exchange between the $^{13}$C and the $^{1}$H nuclear spins, which was prepared at the hot inverse temperature $\beta_2$ [43], is achieved by a sequence of free evolutions under the natural scalar interaction $H_J = (\pi/2)hJ\sigma_z^{\mathrm{H}}\sigma_z^{\mathrm{C}}$ (with $J \approx 215.1$ Hz) between both nuclei and rf pulses [14] (Appendix A). The resulting fully thermalized state is $\rho_2^{\mathrm{eq}} = \exp(-\beta_2 H_2^{C})/Z_2$, with $H_2^C = H_{\mathrm{exp}}^{\mathrm{C}}(\tau)$. 4) Compression: we finally decrease the nuclear spin energy gap back to its initial value $\nu_1$ in time $\tau$ according to $H_{\mathrm{com}}^{\mathrm{C}}(t) = -H_{\mathrm{exp}}^{\mathrm{C}}(\tau - t)$. In both cases, $\beta_i = (k_B T_i)^{-1}$, $(i = 1, 2)$, where $T_i$ is the effective spin temperature defined via the ratio of the populations of excited and ground states (Appendix A), and $k_B$ is the Boltzmann constant. From the point of view of the working medium, the effective cooling and heating in the experiment are indistinguishable from the thermalization with a heat bath (Appendix A).

## 3 Work-heat correlations

Work and heat fluctuations of the quantum heat engine are characterized by a joint distribution $P(W, Q)$, which can be fully determined in the present experiment by a multipoint measurement scheme along the quantum Otto cycle for different driving times $\tau$ (Fig. 1b), since each stroke involves only work or heat. The protocol consists of two projective energy measurements at the beginning ($\Pi_j$) and at the end ($\Pi_k$) of the expansion stroke, as well as two additional projective energy measurements at the beginning ($\Pi_l$) and at the end ($\Pi_m$) of the compression phase. The statistical analysis of the outcomes of each of the three consecutive pairs of measurements yields a set of transition probabilities among the instantaneous energy eigenstates, which we determine experimentally via a Ramsey-like interferometric method described in detail in Refs. [31–34]. The corresponding joint distribution of the total extracted work $W$ and

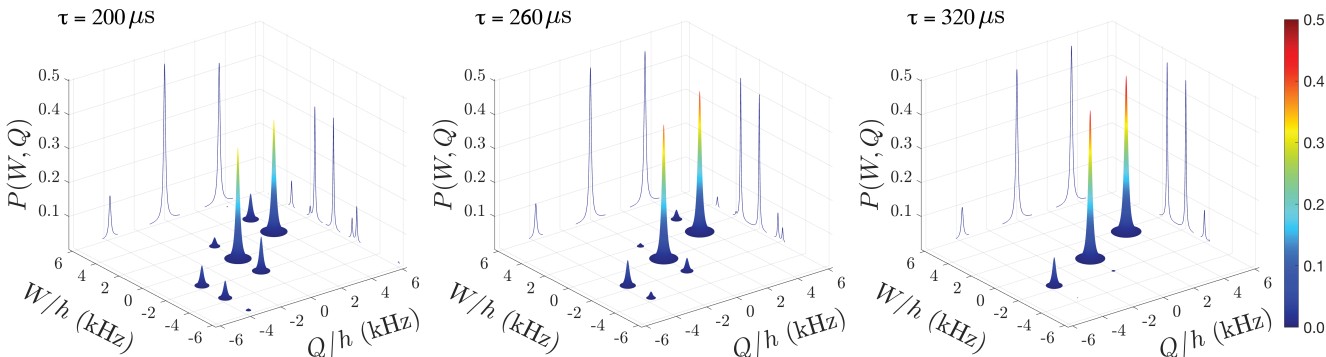

FIG. 2. Joint distribution of work and heat. Distribution $P(W,Q)$, Eq. (1), for three driving times, $\tau = 200$, 260, and 320 $\mu$s in the unitary driving strokes. Experimental data is well fitted by nine Lorentz distributions at the corresponding pair of stochastic values for the extracted work ($W/h = 0, \pm1.6, \pm2.0 \pm 3.6, \pm5.6$ kHz) and heat absorbed from the hot source ($Q/h = 0, \pm3.6$ kHz). Diagonal peaks grow at the expense of off-diagonal ones as the quasiadiabatic regime is approached at $\tau = 320\mu$s.

the absorbed heat $Q$ reads (work is considered positive when extracted from the engine) (Appendix B),

$$P(W,Q) = \sum_{j,k,l,m} \Delta(W,j,k,l,m,\tau,\gamma)\Delta(Q,j,k,l,m,\tau,\gamma)$$
$$\times p_j^0 \, p_{jk}^{\text{exp}} \, p_{kl}^{\text{hea}} \, p_{lm}^{\text{com}}, \tag{1}$$

where $p_j^0 = \exp(-\beta_1 E_j^0)/Z^0$ is the occupation of the cold equilibrium state, with $E_j^0$ the eigenenergies of $H_1^{\text{C}}$. The transition probabilities during expansion, heating and compression are respectively $p_{jk}^{\text{exp}}$, $p_{kl}^{\text{hea}}$ and $p_{ml}^{\text{com}}$. Since the heating stroke leads to a hot equilibrium state, we simply have $p_{kl}^{\text{hea}} = p_l^\tau = \exp(-\beta_2 E_l^\tau)/Z^\tau$, independent of $k$, with $E_j^\tau$ the eigenenergies of $H_2^{\text{C}}$. Occupation probabilities describe the effects of thermal fluctuations, while transition probabilities those of quantum fluctuations and quantum dynamics [44]. For ideal projective measurements, each spectral peak is infinitely sharp ($\gamma = 0$), and energy changes during single strokes are given by differences of energy eigenvalues [45]. In this case, the two functions $\Delta$ associated with work and heat, $X = (W,Q)$, are $\Delta(X,j,k,l,m,\tau,0) = \delta(X - x_{jklm})$, with $w_{jklm} = E_j^0 - E_k^\tau + E_m^\tau - E_l^0$ and $q_{jklm} = E_l^\tau - E_m^\tau$. However, the experimental Ramsey-like interferometric scheme leads to spectral peaks with a finite width $\gamma$, which are well fitted by a Lorentzian distribution, $\Delta(X,j,k,l,m,\tau,\gamma) = 1/\{\pi\gamma[1 + (X - x_{jklm})^2/\gamma^2]\}$ [33, 34]. Taking the marginals of Eq. (1) over either heat or work, we recover the single distributions, $P(W)$ and $P(Q)$, analyzed in Ref. [14].

Examples of experimentally reconstructed bivariate distributions for work and heat are shown in Fig. 2 for three different driving times, for $k_B T_1/h = 1.60 \pm 0.02$ kHz and $k_B T_2/h = 12.21 \pm 0.89$ kHz (results for additional driving times are presented in Appendix C). We observe up to nine discrete Lorentzian peaks, each with a width of about 0.15 kHz. As the driving time increases from $\tau = 200\mu$s to $\tau = 320\mu$s, diagonal peaks grow at the expense of off-diagonal ones. This suggests that work-heat correlations are enhanced as the process becomes more and more adiabatic.

Work-heat correlations within the heat engine cycle are conveniently studied quantitatively with the help of the Pearson coefficient, $\varrho = \text{cov}(W,Q)/\sigma_W\sigma_Q$, defined as the ratio of the covariance and the respective standard deviations [46]. Extracted work and heat are in general (strongly) correlated ($\varrho > 0$) for the quantum Otto engine (Fig. 3a) and correlations oscillate as a function of time, owing to the periodic nature of the driving during expansion and compression steps; dots represent experimental data and the dashed line a numerical simulation (Appendix G). Maximum correlation ($\varrho \simeq 1$) is achieved for quasiadiabatic driving for $\tau = 320\mu$s (Appendix D for a theoretical justification). In this limit, the quantum heat engine satisfies the tight-coupling condition [35–38], which implies that work and heat are proportional. The tight-coupling condition plays a special role in the investigation of the universal properties of heat engines [35–38]. For quantum swap engines, work-heat correlations are controlled by the goodness of the swap operation, irrespective of the degree of adiabaticity [27].

## 4 Efficiency and nonequilibrium entropy production fluctuations

We next move to the analysis of the quantum stochastic efficiency defined as $\eta = W/Q$ for each single realization [47]. This random quantity should not be confused with the deterministic (thermodynamic) efficiency, $\eta_{\text{th}} = \langle W \rangle / \langle Q \rangle$,

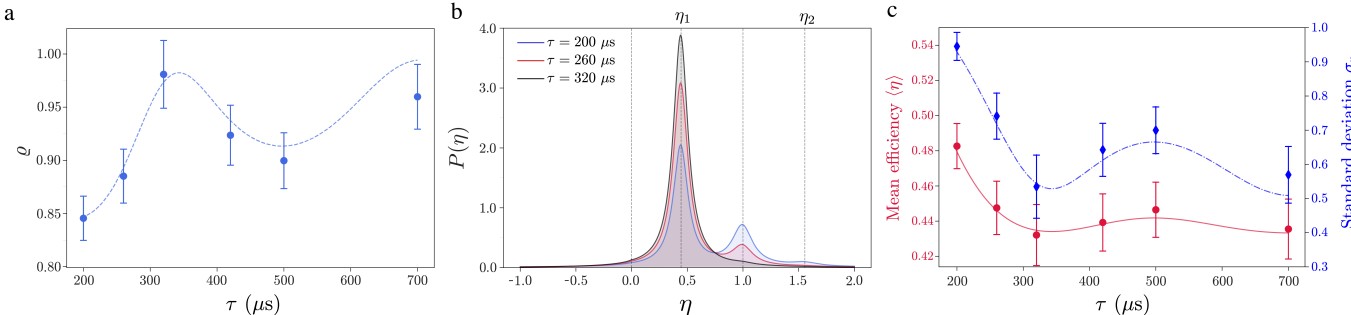

FIG. 3. Work-heat correlations and efficiency distribution. (a) Pearson correlation coefficient $\varrho$ for work and heat as a function of the driving time $\tau$. (b) Efficiency distribution $P(\eta)$, Eq. (2), for three different driving times, displaying two large peaks at 1 and $\eta_1 = \eta_{\text{th}} \approx 0.44$, and two small peaks at 0 and $\eta_2 = 2 - \eta_{\text{th}} \approx 1.56$. (c) Mean efficiency $\langle\eta\rangle$ and corresponding standard deviation $\sigma_\eta$ as a function of the driving time $\tau$. Work and heat are maximally correlated as the tight-coupling condition is approached for $\tau = 320\mu s$. As a result, the stochastic efficiency is sharply peaked around the thermodynamic efficiency $\eta_{\text{th}}$ and the standard deviation, which quantifies fluctuations, strongly decreases. By contrast, the mean random efficiency decreases. Symbols are the experimental data and lines show theoretical predictions.

which is given in terms of averages [1]. The stochastic efficiency $\eta$ corresponds to the efficiency of a single engine along one cycle, whereas the deterministic efficiency $\eta_{\text{th}}$ that of an ensemble of identical engines. In the case of adiabatic driving, the latter reduces to the standard Otto efficiency, $\eta_{\text{Otto}} = 1 - \nu_1/\nu_2$ [29]. The efficiency distribution $P(\eta)$ follows from the joint distribution (1) via integration over all work and heat values,

$$P(\eta) = \iint dW \, dQ \; \delta\left(\eta - \frac{W}{Q}\right) P(W, Q). \tag{2}$$

The corresponding experimental distribution is displayed in Fig. 3b for three different driving times. We identify four Lorentzian-like peaks (Appendix E): two (large) peaks found at 1 and $\eta_1 = \eta_{\text{th}}$, and two (small) peaks located at 0 and $\eta_2 = 2 - \eta_{\text{th}}$. The stochastic efficiency is further seen to take values above 1 and below 0. In the former case, the produced random work is larger than the absorbed stochastic heat, while in the latter case work is added to the machine or heat is given to the hot bath. These results indicate that all values of the stochastic efficiency are possible in a small-scale quantum engine running in finite time, including those forbidden by the macroscopic second law. As the driving time approaches the adiabatic regime ($\tau = 320\mu s$), we observe that the thermodynamic efficiency $\eta_{\text{th}}$ becomes increasingly more likely. In order to examine the properties of the stochastic efficiency $\eta$, we evaluate its mean $\langle\eta\rangle$ and standard deviation $\sigma_\eta$ in the interval $[-5, 5]$ (Fig. 3c). The behavior of the mean efficiency and of its standard deviation as a function of $\tau$ is exactly opposite to that the Pearson coefficient (Fig. 3a): they decrease when the correlations increase, and vice versa, revealing the strong relationship existing between the work-heat correlations and the features of the stochastic efficiency. Surprisingly, the dependence of the mean random efficiency $\langle\eta\rangle$ on $\tau$ is at variance with that of the deterministic efficiency $\eta_{\text{th}}$ (Appendix E). The stochastic efficiency is thus larger for nonadiabatic than for adiabatic driving; this is due to the peaks above $\eta_{\text{th}}$ and, hence, to events violating the macroscopic second law which are more likely for nonadiabatic driving. We also note that the stochastic efficiency tends to the deterministic efficiency $\eta_{\text{th}}$ as the tight-coupling limit is approached. The effects of fluctuations are here significantly suppressed due to the strong work-heat correlation, even though these fluctuations do not vanish [47].

Energy fluctuations in a heat engine cycle are predicted to obey a detailed fluctuation relation of the form [24–27],

$$\frac{P(W, Q)}{P(-W, -Q)} = e^{\Delta\beta Q - \beta_1 W}, \tag{3}$$

where $\Delta\beta = \beta_1 - \beta_2$ and $P(-W, -Q)$ is the joint distribution of measuring $(-W, -Q)$ in the reverse operation of the engine. An integral fluctuation theorem, $\langle\exp(-\Sigma)\rangle = \iint dW \, dQ \; P(W, Q)\exp(-\Sigma) = 1$, for the entropy production $\Sigma = \Delta\beta Q - \beta_1 W$ follows after integration over one cycle [24–27]. The latter expression may be regarded as a nonequilibrium generalization of the Carnot formula, $\langle W\rangle/\langle Q\rangle \leq 1 - T_1/T_2$, which can be derived from it by applying Jensen's inequality [24–27]. Figure 4a displays a verification of the quantum detailed fluctuation relation (3) for $\tau = 200 \mu s$ (Appendix F for other driving times). We witness very good agreement between the experimental values of $\ln[P(W, Q)/P(-W, -Q)]$ (red dots) and the predictions of Eq. (3) indicated by the (blue) plane $z = \Sigma$, the $z$-axis being vertical. The excellent agreement is further confirmed by the projections onto the work/heat planes, as seen in

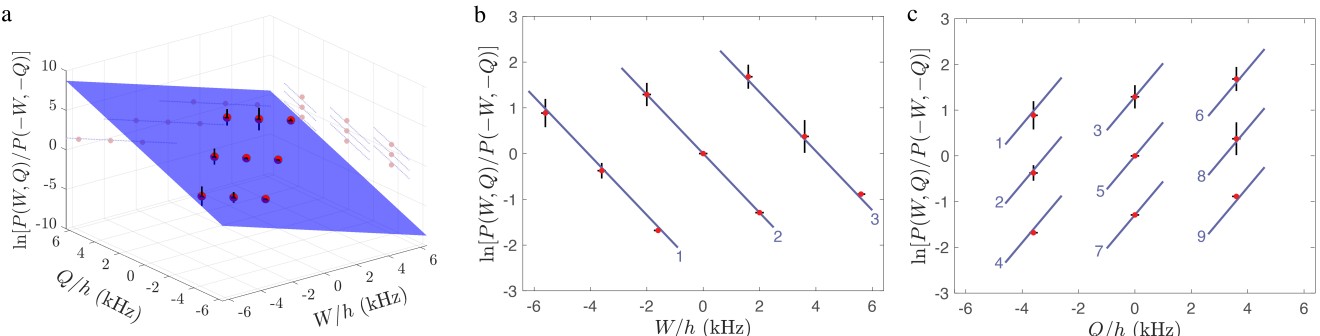

FIG. 4. Nonequilibrium quantum fluctuation relation. (a) Experimental verification of the detailed fluctuation relation (3) for the quantum Otto cycle: the values of $\ln\left[P(W,Q)/P(-W,-Q)\right]$ (red dots) should lie within the (blue) plane defined by the total entropy production $z = \Sigma = \Delta\beta Q - \beta_1 W$; the dashed blue lines show the respective projections of the plane on the work and heat axes. (b) Confirmation of the joint fluctuation relation in the work domain: the experimental values of $\ln[P(W,Q)/P(-W,-Q)]$ (red dots) should intercept the three (blue) lines corresponding to the projection of the plane defined by $z = \Sigma$, for each possible value of heat from the hot source, $Q/h = -3.6, 0.0, 3.6$ kHz, associated to the curves identified as $1, 2, 3$, respectively. (c) Confirmation of the joint fluctuation relation in the heat domain: the experimental values of $\ln[P(W,Q)/P(-W,-Q)]$ (red dots) should intercept the nine (blue) lines corresponding to the projection of the plane defined by $z = \Sigma$, for each possible value of work, $W/h = -5.6, -3.6, -2.0, -1.6, 0.0, 1.6, 2.0, 3.6, 5.6$ kHz, associated to the curves $1, \cdots, 9$, respectively. The data corresponds to the driving time $\tau = 200$ $\mu$s.

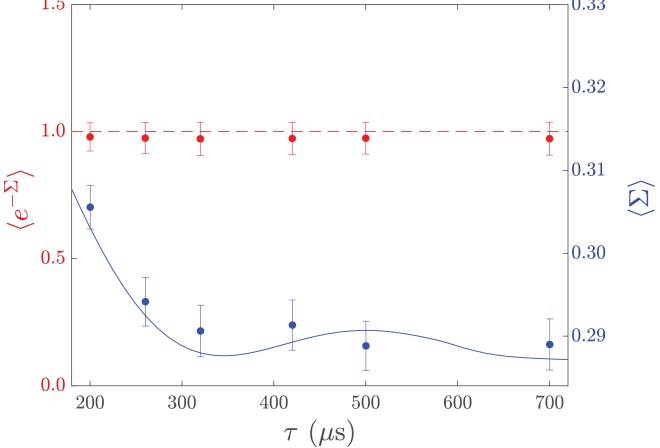

FIG. 5. Quantum integral fluctuation theorem. Experimental confirmation of the integral fluctuation theorem, $\langle \exp(-\Sigma) \rangle = 1$, (red dots) and average entropy production $\langle \Sigma \rangle$ (blue dots) for the thermal cycle as a function of the driving time $\tau$. Irreversible losses are minimal when the tight-coupling condition is approached for $\tau = 320\mu$s.

Figs. 4b and 4c. A verification of the integral fluctuation theorem, $\langle \exp(-\Sigma) \rangle = 1$, is moreover shown in Fig. 5 as a function of the driving time, together with the average entropy production $\langle \Sigma \rangle$, which characterizes irreversible losses within the cycle. Since the two driving Hamiltonians, $H_{\exp}^{C}(t)$ and $H_{\mathrm{com}}^{C}(t)$, do not commute at different times, the heat engine exhibits internal friction associated with nonadiabatic transitions between the instantaneous eigenstates of the $^{13}$C nuclear spin [39–41]. This purely quantum friction mechanism is the source of irreversibility in the quantum Otto cycle, depending on the driving speed: the mean entropy production $\langle \Sigma \rangle$ decreases as the adiabatic regime is approached (Fig. 4b), and vice versa. We also note a marked connection between quantum friction (Fig. 5) and work-heat correlations (Fig. 3a), which has not been acknowledged before.

## 5 Conclusions

In conclusion, we have performed an experimental study of the work-heat correlations and their strong impact on both efficiency and entropy production statistics of a quantum heat engine. We have shown that the tight-coupling condition, corresponding to maximum work-heat correlation, can be reached for finite-time quasiadiabatic driving.

In this regime, the stochastic efficiency reduces to the deterministic efficiency, and both thermal and quantum fluctuations are notably suppressed. We have additionally observed that deterministic and stochastic efficiencies display opposite behavior, due to random events violating the macroscopic second law. We have finally confirmed nonequilibrium generalizations of the Carnot formula in the form of bivariate (detailed and integral) fluctuation relations for work and heat, and analyzed the effect of quantum friction on the total entropy production. Our findings provide unique insight into the nonequilibrium fluctuating properties of small quantum thermal machines, and offer possibilities to directly control them.

## Acknowledgements

We acknowledge financial support from the Federal University of ABC (UFABC), the Brazilian National Council for Scientific and Technological Development (CNPq), the Brazilian Federal Agency for Support and Evaluation of Graduate Education (CAPES), the São Paulo Research Foundation (FAPESP) (Grant number 19/04184-5) and the German Science Foundation (DFG) (Project FOR 2724). This research was performed as part of the National Institute for Science and Technology of Quantum Information (CNPq, INCT-IQ 465469/2014-0). We also thank the Multiuser Central Facilities of UFABC.

## Appendices

## A Experimental Protocols

### A1 Thermal states initialization

Spatial average techniques [30, 33, 34, 43] were used to initialize the engine states, which are local pseudo-thermal states encoded in the $^1$H and $^{13}$C nuclei. We present in Table I the populations and the respective local spin temperatures in the eigenbasis of Hamiltonians $\mathcal{H}_0^{\mathrm{H}}$ and $\mathcal{H}_0^{\mathrm{C}}$.

The effective spin temperatures $T_i$ are determined via the ratio of the (Boltzmann distributed) populations of excited ($p_1^{H(C)}$) and ground ($p_0^{H(C)}$) states of the respective spin-1/2 systems [30, 33, 34]:

$$k_B T_{2(1)} = h\nu_{2(1)}[\ln(p_0^{H(C)}/p_1^{H(C)})]^{-1}, \tag{4}$$

where $\nu_i$ are the corresponding frequencies.

| $^1$H nucleus | $p_0^{\mathrm{H}}$ | $p_1^{\mathrm{H}}$ | $k_B T_2$ (peV) |
|---|---|---|---|
| | $0.67 \pm 0.01$ | $0.33 \pm 0.01$ | $21.5 \pm 0.4$ |
| $^{13}$C nucleus | $p_0^{\mathrm{C}}$ | $p_1^{\mathrm{C}}$ | $k_B T_1$ (peV) |
| | $0.78 \pm 0.01$ | $0.22 \pm 0.01$ | $6.6 \pm 0.1$ |

TABLE I. Populations and spin temperatures of the initial states of the $^1$H and $^{13}$C nuclei. The corresponding off-diagonal elements are zero within the measurement errors.

### A2 Compression and expansion protocols

The energy gap compression and expansion protocols are implemented with a time-modulated amplitude and phase transverse rf-pulse on resonance with the $^{13}$C nuclear spin in order to produce effectively the time-dependent driving Hamiltonian $\mathcal{H}^C(\tau)$ described in the main text. The intensities of the transverse field at the beginning and end of the driving protocol were properly calibrated in order to have the associated frequencies given in the main text. The duration of the modulated traverse pulse was varied from 100 $\mu$s to 700 $\mu$s in different implementations of the quantum heat engine cycle.

### A3 Heating protocol

The thermalization process used to heat the $^{13}$C nuclear spin of the quantum Otto engine during the second stroke has the local effect of a linear non-unitary map $\varepsilon(\rho_j) = \mathrm{Tr}_{k \neq j} [\mathcal{U}_\tau (\rho_H^0 \otimes \rho_C^0) \mathcal{U}_\tau^\dagger]$, with $(j, k) = (H, C)$. It is represented

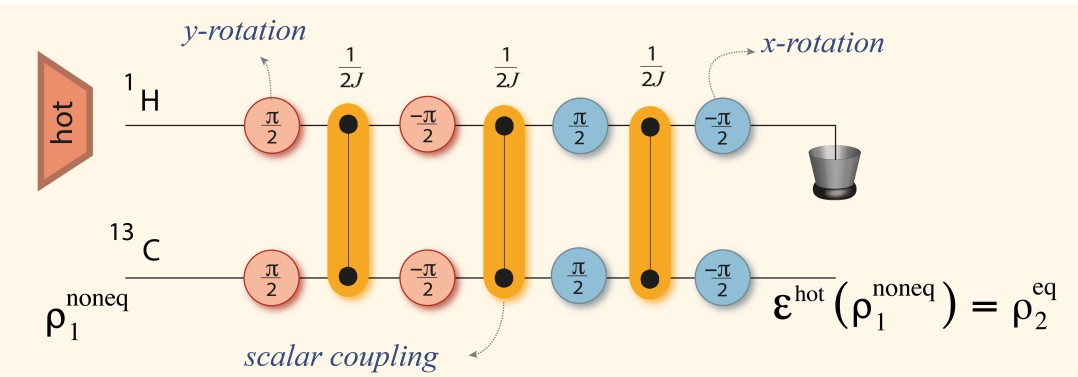

FIG. 6. NMR pulse sequence used in the heat exchange protocol. The outcome of this sequence (which takes about 7 ms) is an effective full thermalization described by a completely positive trace preserving (CPTP) map on reduced density operator of the carbon nucleus, $\varepsilon^{\mathrm{hot}} : \rho_1^{\mathrm{noneq}} \to e^{-\beta_2 H_2^{\mathrm{C}}}/Z_2$, leading it to an equilibrium state at the hot inverse temperature $\beta_2$. Orange connections represent free evolutions under the scalar interaction during the time displayed above the symbol. Blue (red) circles stand for $x$ ($y$) rotations by the displayed angle implemented by transverse rf pulses.

by the following set of maps [43]

$$\varepsilon(\rho_j) = \sum_{\ell=1}^{4} K_\ell \rho_j^0 K_\ell^\dagger, \tag{5}$$

with the Kraus operators

$$K_1 = \sqrt{1-p} \begin{pmatrix} 1 & 0 \\ 0 & 0 \end{pmatrix}, K_2 = \sqrt{p} \begin{pmatrix} 0 & 0 \\ 0 & 1 \end{pmatrix} \tag{6}$$

$$K_3 = \sqrt{1-p} \begin{pmatrix} 0 & 1 \\ 0 & 0 \end{pmatrix}, K_4 = \sqrt{p} \begin{pmatrix} 0 & 0 \\ -1 & 0 \end{pmatrix}. \tag{7}$$

The parameter $p$ denotes the population of the excited state in the Hydrogen nucleus. The above Kraus operators correspond to the generalized amplitude damping of a spin-1/2 system. From a local point of view, the map thus implements complete thermalization. The NMR pulse sequence used in the heat exchange protocol in the experiment is shown in Fig. 6, were the Hydrogen nucleus is used as a heat bus.

## B  Joint distribution for work and heat - theory

The joint distribution for the total work $W$ and the absorbed heat $Q$ may be determined by performing energy measurements on the engine at the beginning and at the end of the expansion, heating and compression strokes [47, 48], as depicted in Fig. 1 of the main text. We first consider the case of ideal projective measurements. By performing projective energy measurements at the beginning and at the end of the expansion step, the distribution of the expansion work $W_2$ reads [45],

$$P(W_2) = \sum_{j,k} \delta \left[ W_2 - (E_k^\tau - E_j^0) \right] p_{jk}^{\mathrm{exp}} p_j^0, \tag{8}$$

where $E_j^0$ and $E_k^\tau$ are the respective initial and final energy eigenvalues, $p_j^0 = \exp(-\beta_1 E_j^0)/Z^0$ is the initial thermal occupation probability with partition function $Z^0$ and $p_{jk}^{\mathrm{exp}} = |\langle j| U_{\mathrm{exp}}(\tau) |k\rangle|^2$ denotes the transition probability between the instantaneous eigenstates $|j\rangle$ and $|k\rangle$ in time $\tau$ with the corresponding unitary $U_{\mathrm{exp}}$.

Similarly, the probability density of the heat $Q = Q_3$ during the following heating step, given the expansion work $W_2$, is equal to the conditional distribution [49],

$$P(Q|W_2) = \sum_{i,l} \delta \left[ Q - (E_l^\tau - E_i^\tau) \right] p_{il}^{\mathrm{hea}} p_i^\tau, \tag{9}$$

where the occupation probability at time $\tau$ is $p_i^\tau = \delta_{ki}$ when the system is in eigenstate $|k\rangle$ after the second projective energy measurement.

| History | stroke 1 | stroke 2 | stroke 3 | stroke 4 | $W/h \pm 0.15$ (kHz) | $Q/h \pm 0.15$ (kHz) |
|---|---|---|---|---|---|---|
| 1 | $\lvert\Psi_-^1\rangle$ | $\lvert\Psi_-^2\rangle$ | $\lvert\Psi_-^2\rangle$ | $\lvert\Psi_-^1\rangle$ | 0 | 0 |
| 2 | $\lvert\Psi_-^1\rangle$ | $\lvert\Psi_-^2\rangle$ | $\lvert\Psi_-^2\rangle$ | $\lvert\Psi_+^1\rangle$ | -2.0 | 0 |
| 3 | $\lvert\Psi_-^1\rangle$ | $\lvert\Psi_-^2\rangle$ | $\lvert\Psi_+^2\rangle$ | $\lvert\Psi_-^1\rangle$ | 3.6 | 3.6 |
| 4 | $\lvert\Psi_-^1\rangle$ | $\lvert\Psi_-^2\rangle$ | $\lvert\Psi_+^2\rangle$ | $\lvert\Psi_+^1\rangle$ | 1.6 | 3.6 |
| 5 | $\lvert\Psi_-^1\rangle$ | $\lvert\Psi_+^2\rangle$ | $\lvert\Psi_-^2\rangle$ | $\lvert\Psi_-^1\rangle$ | -3.6 | -3.6 |
| 6 | $\lvert\Psi_-^1\rangle$ | $\lvert\Psi_+^2\rangle$ | $\lvert\Psi_-^2\rangle$ | $\lvert\Psi_+^1\rangle$ | -5.6 | -3.6 |
| 7 | $\lvert\Psi_-^1\rangle$ | $\lvert\Psi_+^2\rangle$ | $\lvert\Psi_+^2\rangle$ | $\lvert\Psi_-^1\rangle$ | 0 | 0 |
| 8 | $\lvert\Psi_-^1\rangle$ | $\lvert\Psi_+^2\rangle$ | $\lvert\Psi_+^2\rangle$ | $\lvert\Psi_+^1\rangle$ | -2.0 | 0 |
| 9 | $\lvert\Psi_+^1\rangle$ | $\lvert\Psi_-^2\rangle$ | $\lvert\Psi_-^2\rangle$ | $\lvert\Psi_-^1\rangle$ | 2.0 | 0 |
| 10 | $\lvert\Psi_+^1\rangle$ | $\lvert\Psi_-^2\rangle$ | $\lvert\Psi_-^2\rangle$ | $\lvert\Psi_+^1\rangle$ | 0 | 0 |
| 11 | $\lvert\Psi_+^1\rangle$ | $\lvert\Psi_-^2\rangle$ | $\lvert\Psi_+^2\rangle$ | $\lvert\Psi_-^1\rangle$ | 5.6 | 3.6 |
| 12 | $\lvert\Psi_+^1\rangle$ | $\lvert\Psi_-^2\rangle$ | $\lvert\Psi_+^2\rangle$ | $\lvert\Psi_+^1\rangle$ | 3.6 | 3.6 |
| 13 | $\lvert\Psi_+^1\rangle$ | $\lvert\Psi_+^2\rangle$ | $\lvert\Psi_-^2\rangle$ | $\lvert\Psi_-^1\rangle$ | -1.6 | -3.6 |
| 14 | $\lvert\Psi_+^1\rangle$ | $\lvert\Psi_+^2\rangle$ | $\lvert\Psi_-^2\rangle$ | $\lvert\Psi_+^1\rangle$ | -3.6 | -3.6 |
| 15 | $\lvert\Psi_+^1\rangle$ | $\lvert\Psi_+^2\rangle$ | $\lvert\Psi_+^2\rangle$ | $\lvert\Psi_-^1\rangle$ | 2.0 | 0 |
| 16 | $\lvert\Psi_+^1\rangle$ | $\lvert\Psi_+^2\rangle$ | $\lvert\Psi_+^2\rangle$ | $\lvert\Psi_+^1\rangle$ | 0 | 0 |

TABLE II. All transition histories between the instantaneous eigenstates $\lvert\Psi_\pm^i\rangle$ ($i = 1, 2$) for each stroke of the heat engine, together with the corresponding values of work and heat.

The quantum work distribution for compression, given the expansion work $W_2$ and the heat $Q$, is additionally,

$$P(W_4|Q, W_2) = \sum_{r,m} \delta\left[W_4 - (E_m^0 - E_r^\tau)\right] p_{rm}^{\text{com}} p_r^\tau, \tag{10}$$

with the occupation probability $p_r^\tau = \delta_{rl}$ when the system is in eigenstate $\lvert l\rangle$ after the third projective energy measurement. The transition probability $p_{rm}^{\text{com}} = \lvert\langle r\rvert U_{\text{com}}(\tau)\lvert m\rangle\rvert^2$ is fully specified by the unitary time evolution operator for compression $U_{\text{com}}$.

The joint probability of having certain values of $W_4$, $Q$ and $W_2$ during a cycle of the quantum engine now follows from the chain rule for conditional probabilities, $P(W_4, Q, W_2) = P(W_4|Q, W_2)P(Q|W_2)P(W_2)$ [50]. Using Eqs. (8), (9) and (10), we find,

$$\begin{aligned}
P(W_2, Q, W_4) = \sum_{j,k,l,m} & \delta\left[W_2 - (E_k^\tau - E_j^0)\right] \\
& \times \delta\left[Q - (E_l^\tau - E_m^\tau)\right] \delta\left[W_4 - (E_m^0 - E_l^\tau)\right] \\
& \times p_j^0 \, p_{jk}^{\text{exp}} \, p_{kl}^{\text{hea}} \, p_{lm}^{\text{com}}.
\end{aligned} \tag{11}$$

Introducing the total extracted work $W = -(W_2 + W_4)$ work and integrating over all work values $W_2$ and $W_4$, the joint distribution for work and heat is given by,

$$P(W, Q) = \int dW_2 dW_4 \, \delta[W + (W_2 + W_4)]P(W_2, Q, W_4). \tag{12}$$

Using the explicit expression (11), we finally obtain,

$$\begin{aligned}
P(W, Q) = \sum_{j,k,l,m} & \Delta(W, j, k, l, m, \tau, \gamma)\Delta(Q, j, k, l, m, \tau, \gamma) \\
& \times p_j^0 \, p_{jk}^{\text{exp}} \, p_{kl}^{\text{hea}} \, p_{lm}^{\text{com}}.
\end{aligned} \tag{13}$$

For ideal projective measurements, each spectral peak is infinitely sharp ($\gamma = 0$) and the two functions $\Delta$ associated with work and heat, $X = (W, Q)$, are simply Dirac peaks, $\Delta(X, j, k, l, m, \tau, 0) = \delta(X - x_{jklm})$, with $w_{jklm} = E_j^0 - E_k^\tau + E_m^\tau - E_l^0$ and $q_{jklm} = E_l^\tau - E_m^\tau$.

In the experiment, each pair of energy measurements is effectively implemented using a Ramsey-like interferometric scheme [31–34]. In this case, spectral peaks have a finite width $\gamma$ and are well fitted by a Lorentzian distribution, $\Delta(X, j, k, l, m, \tau, \gamma) = 1/\{\pi\gamma[1 + (X - x_{jklm})^2/\gamma^2]\}$ [33, 34]. This is the form we consider in the present experiment.

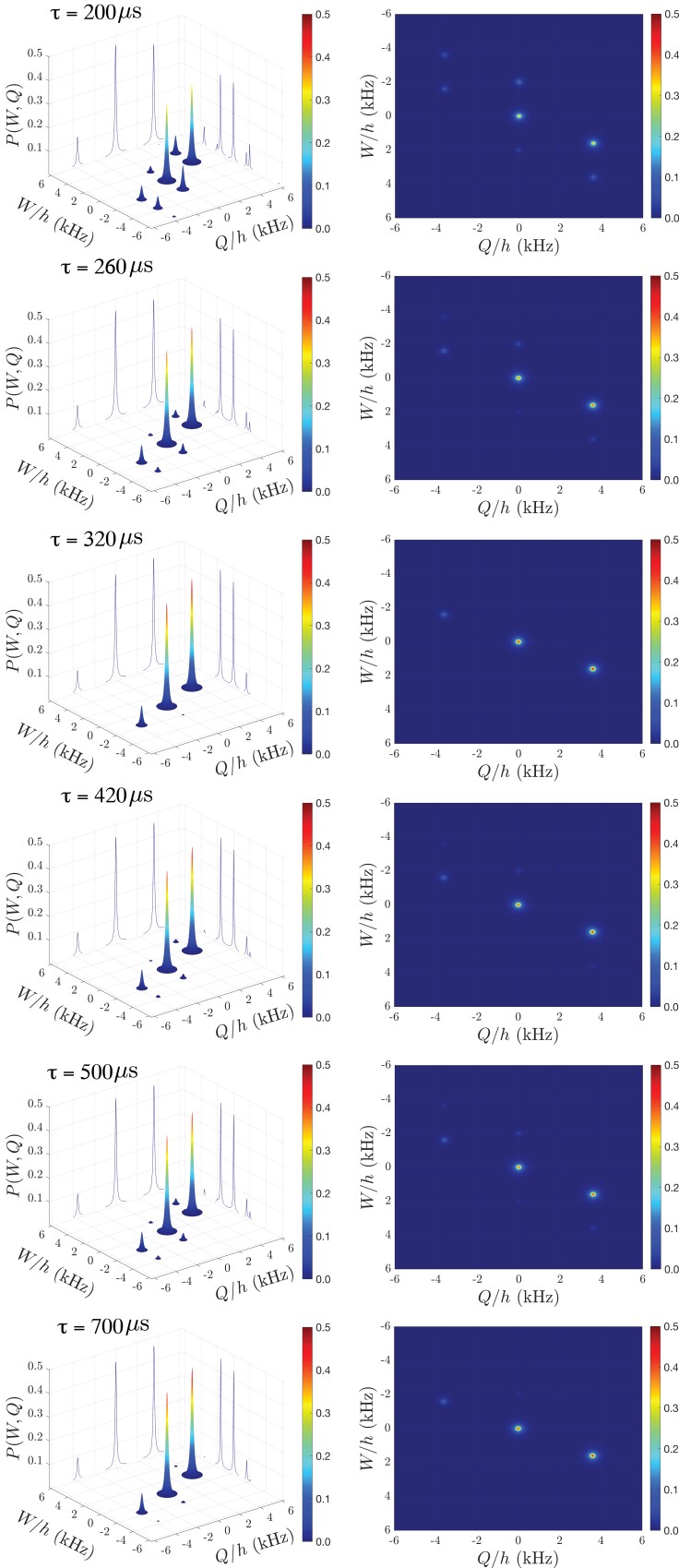

FIG. 7. Reconstructed joint probability distribution for work and heat $P(W, Q)$, Eq. (13), for the following values of the driving times, $\tau = 200$, 260, 320, 260, 420, 500, and 700 $\mu$s (left), together with the corresponding density plots (right).

## C Joint distribution for work and heat - experiment

We denote the instantaneous energy eigenstates of the two-level system with energy gap $h\nu_i$ ($i = 1, 2$) as $|\Psi_\pm^i\rangle$. The corresponding transition probabilities during expansion and compression strokes are accordingly given by

$$|\langle\Psi_-^1|\, U\,|\Psi_-^2\rangle|^2 = |\langle\Psi_+^1|\, U\,|\Psi_+^2\rangle|^2 = 1 - \xi, \tag{14}$$

when there is no transition between states, and by

$$|\langle\Psi_-^1|\, U\,|\Psi_+^2\rangle|^2 = |\langle\Psi_+^1|\, U\,|\Psi_-^2\rangle|^2 = \xi. \tag{15}$$

when there is a change of state. The operator $U$ stands for the expansion or compression unitary. Adiabatic driving corresponds to $\xi = 0$.

Table II presents all the sixteen possible combinations for energy transitions of the quantum Otto heat engine during one cycle, together with the respective values of the extracted random values of work and heat.

The reconstructed joint distributions $P(W, Q)$ are displayed in Fig. 7 for the following values of the driving time, $\tau = 200, 260, 320, 260, 420, 500,$ and $700\,\mu$s.

## D Maximum correlations for adiabatic driving

We next show that work and heat are maximally correlated in the adiabatic regime. To that end, we compute the Pearson correlation coefficient for generic adiabatic scale invariant quantum Otto heat engines with Hamiltonian $H_t = \boldsymbol{p}^2/2m + U(\boldsymbol{x}, \varepsilon_\tau)$, where $U(\boldsymbol{x}, \varepsilon_\tau) = U_0(\boldsymbol{x}/\varepsilon_\tau)/\varepsilon_\tau^2$ with scaling parameter $\varepsilon_\tau$. These Hamiltonians describe a wide class of single-particle, many-body and nonlinear systems with scale-invariant spectra, $E_j^\tau = E_j^0/\varepsilon_\tau^2$ [52]. Two-level systems (as well as harmonic oscillators) satisfy this property. For simplicity, we consider the case of ideal projective measurements ($\gamma = 0$). In the adiabatic regime, we have $|\langle j|\, U_{\exp}\,|k\rangle|^2 = \delta_{jk}$ and $|\langle l|\, U_{\text{com}}\,|m\rangle|^2 = \delta_{lm}$. The joint distribution (13) for work and heat then reduces to,

$$P_{\text{ad}}(W, Q) = \sum_{j,l} \delta\left[W - (1 - \varepsilon_\tau^{-2})\left(E_l^0 - E_j^0\right)\right]$$

$$\times \delta\left[Q - \left(E_l^0 - E_j^0\right)\varepsilon_\tau^{-2}\right] \times \frac{e^{-\beta_c E_j^0 - \beta_h E_l^0/\varepsilon_\tau^2}}{Z_0 Z_\tau}. \tag{16}$$

The Pearson correlation coefficient follows accordingly as,

$$\rho = \frac{\text{cov}(W, Q)}{\sigma_W \sigma_Q} = \frac{\langle WQ\rangle - \langle W\rangle\langle Q\rangle}{(\langle W^2\rangle - \langle W\rangle^2)(\langle Q^2\rangle - \langle Q\rangle^2)}$$

$$= \frac{\left(1 - \varepsilon_\tau^{-2}\right)}{|\left(1 - \varepsilon_\tau^{-2}\right)|} = \pm 1. \tag{17}$$

Work-heat correlations are therefore always maximal in the adiabatic regime. Quantum heat engine conditions further imply that

$$\langle Q\rangle = \varepsilon_\tau^{-2} \sum_{j\neq l} \frac{e^{-\beta_c E_j^0 - \beta_h E_l^0/\varepsilon_\tau^2}}{Z_0 Z_\tau}\left(E_l^0 - E_j^0\right) \geq 0 \tag{18}$$

$$\langle W\rangle = \left(1 - \varepsilon_\tau^{-2}\right) \sum_{j\neq l} \frac{e^{-\beta_c E_j^0 - \beta_h E_l^0/\varepsilon_\tau^2}}{Z_0 Z_\tau}\left(E_l^0 - E_j^0\right) \geq 0.$$

As a result, $\left(1 - \varepsilon_\tau^{-2}\right) \geq 0$, and work output and heat input are perfectly correlated for adiabatic driving, $\rho = 1$. Adiabatic scale invariant quantum Otto engines therefore obey the tight-coupling condition.

## E Efficiency distribution

The stochastic efficiency is defined as $\eta = W/Q$. Its distribution may be obtained from the joint distribution $P(W, Q)$, Eq. (13), by integrating over $W$ and $Q$, as

$$P(\eta) = \int dQ\, dW\, \delta\left(\eta - \frac{W}{Q}\right) P(W, Q)$$

$$= \sum_{j,k,l,m} p_j^0\, p_{jk}^{\exp}\, p_{kl}^{\text{hea}}\, p_{lm}^{\text{com}}\, L(w, q, \gamma, \eta) \tag{19}$$

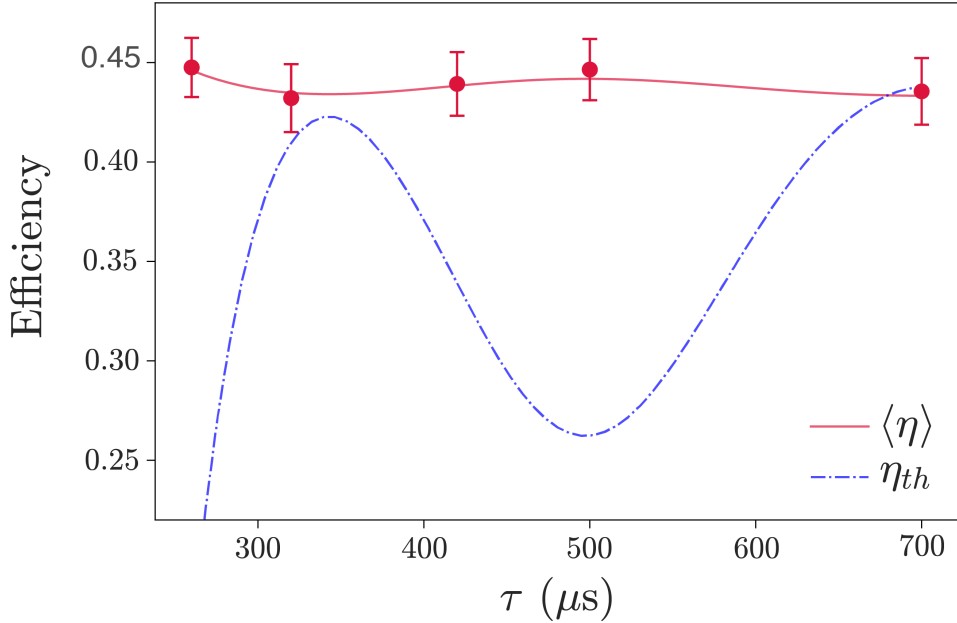

FIG. 8. Comparison of the microscopic mean efficiency $\langle\eta\rangle = \langle W/Q\rangle$ (experimental red dots) and the macroscopic efficiency $\eta_{\mathrm{th}} = \langle W\rangle/\langle Q\rangle$ (simulated blue line) as a function of the driving $\tau$. The macroscopic efficiency increases as the adiabatic regime is approached, while the microscopic average efficiency decreases.

with Lorentz-like peaks,

$$
\begin{aligned}
L(w, q, \gamma, \eta) = {} & \frac{\gamma}{\pi^2 \left(\gamma^2(\eta-1)^2 + \eta^2 q^2 + 2\eta q w + w^2\right)\left(\gamma^2(\eta+1)^2 + \eta^2 q^2 + 2\eta q w + w^2\right)} \\
& \times \Big\{ \gamma\left(-\gamma^2 + \eta^2\left(\gamma^2 + q^2\right) - w^2\right)\left(\log\left(\eta^2\right) + \log\left(\gamma^2 + q^2\right) - \log\left(\gamma^2 + w^2\right)\right) \\
& + 2\tan^{-1}\left(\frac{q}{\gamma}\right)\left(\eta^2 q\left(\gamma^2 + q^2\right) + 2\eta w\left(\gamma^2 + q^2\right) + q\left(\gamma^2 + w^2\right)\right) \\
& + 2\tan^{-1}\left(\frac{w}{\gamma}\right)\left(\eta^2 w\left(\gamma^2 + q^2\right) + 2\eta q\left(\gamma^2 + w^2\right) + w\left(\gamma^2 + w^2\right)\right) \Big\}
\end{aligned}
\tag{20}
$$

where we have dropped the indices of $w$ and $q$ for better readability.

A comparison of the microscopic mean efficiency $\langle\eta\rangle = \langle W/Q\rangle$ and the macroscopic efficiency $\eta_{\mathrm{th}} = \langle W\rangle/\langle Q\rangle$ is displayed in Fig. 8 as a function of the driving time $\tau$. The macroscopic efficiency $\eta_{\mathrm{th}}$ (simulated blue line) increases as the adiabatic regime is approached and irreversible losses induced by quantum friction are reduced. By contrast, the microscopic mean efficiency $\langle\eta\rangle$ (experimental red dots) decreases near the adiabatic regime. It is hence larger for nonadiabatic driving. This counterintuitive behavior is due to the presence of peaks above $\eta_{\mathrm{th}}$ and, thus, to random events that violate the macroscopic second law law of thermodynamics.

## F Detailed fluctuation relation

A test of the detailed quantum fluctuation relation,

$$
\frac{P(W, Q)}{P(-W, -Q)} = e^{-\Delta\beta Q - \beta_1 W}.
\tag{21}
$$

was presented in the main text for the driving time $\tau = 200\,\mu$s. Figure 9 exhibits similar tests for $\tau = 200$, 260, 260, 320, 420, 500, and 700 $\mu$s, showing that the fluctuation theorem for work and heat is obeyed for all the driving times realized in the experiment. Projections onto the respective work and heat planes are shown in Fig. 10 for $\tau = 260$.

## G Numerical simulations

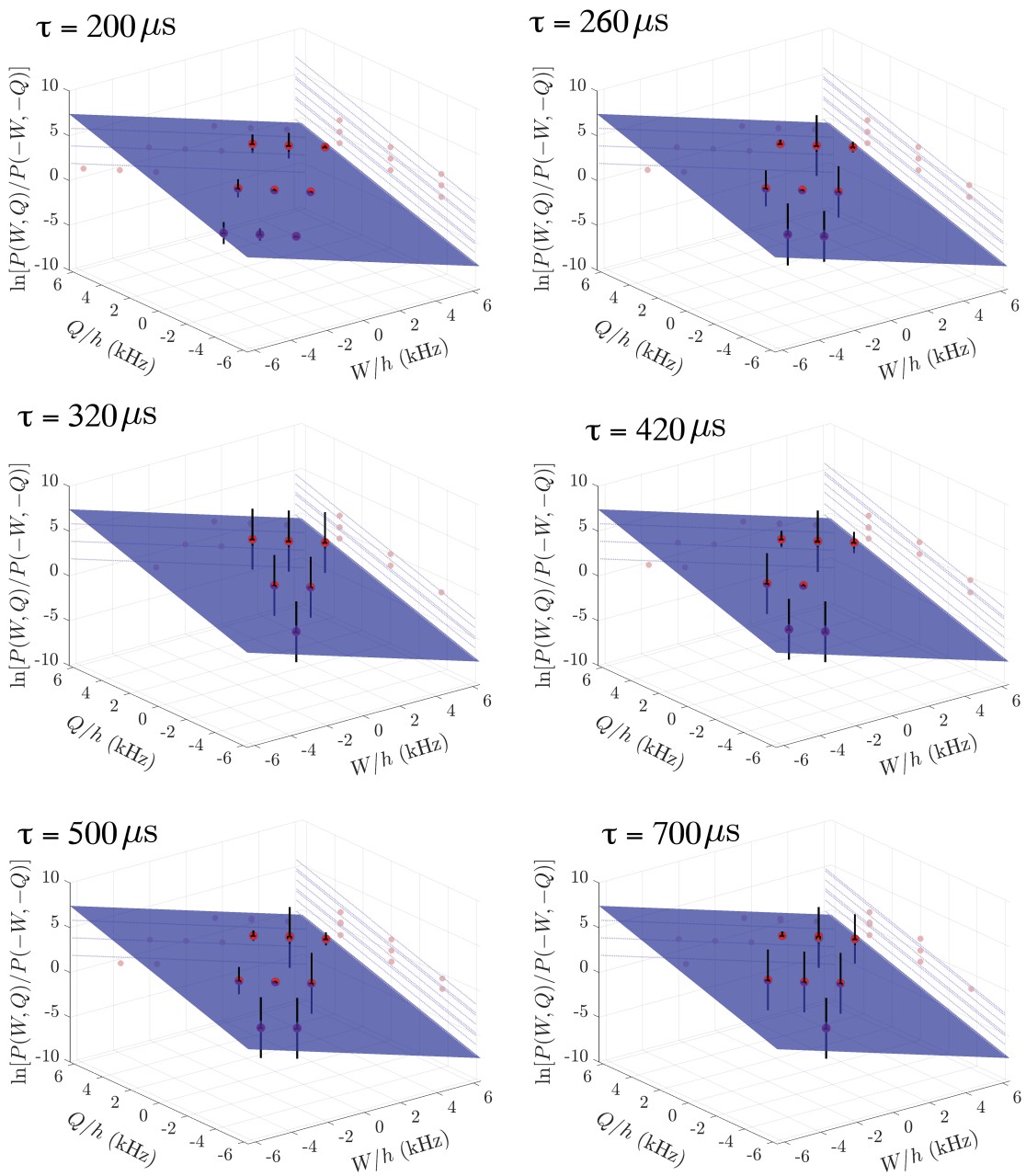

FIG. 9. Experimental verification of the detailed quantum fluctuation relation for work and heat for the following values of the driving time, $\tau = 200$, 260, 320, 260, 420, 500, and 700 $\mu$s.

The numerical simulation of the experiment was implemented using a python-based code (in-house developed) and QuTiP (the Quantum Toolbox in Python) package [53]. We effectively simulated the finite-time quantum Otto cycle described in the main text with the thermalization strokes being solved using the theoretical thermalization of a qubit with a Markovian thermal reservoir in terms of the Bloch vector components [54]. The time-dependent unitary dynamics of the energy gap expansion and compression strokes were solved numerically. In order to obtain the theoretical transition probability, we ran the simulation from $\tau = 100\mu s$ to $\tau = 700\mu s$ considering 50 time steps, which was sufficient to generate smooth curves for the theoretical quantities and for the confirmation of the quantum fluctuation relations.

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

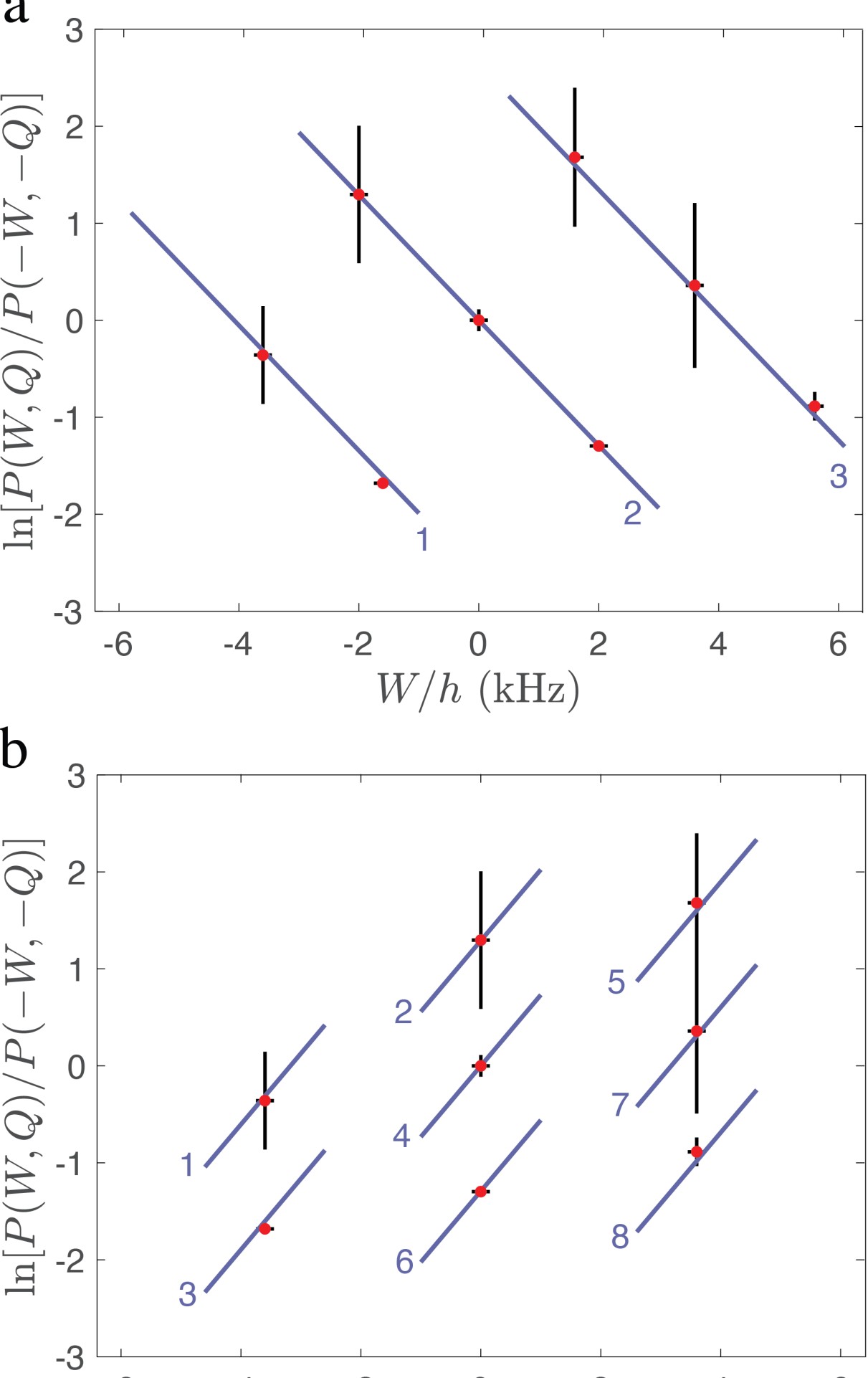