# Peer review of "Nonequilibrium fluctuations of a quantum heat engine"

_SciPost Physics_

## Round 1 · Referee Report · Anonymous (Referee 1) · 2023-10-9

Report

In the present manuscript, the authors have experimentally studied the correlations and fluctuations of work and heat in the finite-time quantum Otto cycle using a spin 1/2 in the liquid-state NMR. In addition, they have tested the fluctuation relation for heat engines.

In my opinion, the problem itself studied in the manuscript, fluctuations of work and heat in quantum heat engines, is interesting and important in quantum thermodynamics. However, I found that there is a crucial problem in their analysis as I shall explain in detail below. Therefore, I cannot recommend publication of this manuscript unless the authors resolve this problem. Regarding the acceptance criteria, I think the only relevant one that can be considered is, "Detail a groundbreaking theoretical/experimental/computational discovery". Considering this criterion, I am skeptical that the present manuscript meets this criterion because of the above mentioned problem in their analysis.

In finite-time quantum heat engines, there are not only diabatic transitions in the populations (i.e., diagonal elements in the instantaneous energy basis) but also generation of coherence (i.e., off-diagonal elements in the instantaneous energy basis) due to the non-quasistatic driving. Such coherence generated by the finite-time operation has important consequences in the statistics of work and heat in quantum heat engines. However, in the analysis of the present study, coherence generated in the driving strokes is artificially destroyed by the projective energy measurements performed at the end of each stroke. Therefore, work and heat statistics obtained in their analysis is not the correct ones in the finite-time quantum Otto cycle. Rather, what they have measured are work and heat statistics of another cycle consisting of two driving strokes, two isochoric strokes, and energy measurements performed between two consecutive strokes. Even though thermalization in the isochoric strokes is perfect and thus the coherence disappears after each isochoric stroke, we can see that, for example, the energy measurement performed at the beginning of the heating stroke, which destroys the coherence generated in the preceding driving stroke (so that the von Neumann entropy of the system is also changed) affects the fluctuation of Q.

Other points

  1. Fluctuation of efficiency has also been studied based on another measure, which shows some universal bound, in Saryal et al., PRL 127, 190603 (2021) and Ito et al., arXiv:1910.08096. I think that these results should also be referred to in the introduction.

  2. The terms "expansion" and "compression" used in the manuscript are confusing. If I understand correctly, "expansion" and "compression" in the manuscript actually means "energy gap expansion" and "energy gap compression". In other references, stroke 2 is usually called compression stroke and stroke 4 as expansion stroke unlike the present manuscript. Therefore, it is better to explicitly write "energy gap expansion" or "energy gap compression" (i.e., better to avoid simply writing "expansion" or "compression" without the word "energy gap").

(End of the report)

Attachment

---

## Editorial Decision

awaiting_resubmission